# Present and Future Applications of Digital PCR in Infectious Diseases Diagnosis

**DOI:** 10.3390/diagnostics14090931

**Published:** 2024-04-29

**Authors:** Laura Sancha Dominguez, Ana Cotos Suárez, María Sánchez Ledesma, Juan Luis Muñoz Bellido

**Affiliations:** 1Department of Microbiology, Hospital Universitario de Salamanca, 37007 Salamanca, Spain; lsancha@saludcastillayleon.es (L.S.D.); acotos@saludcastillayleon.es (A.C.S.); 2Research Group IIMD-16, Institute for Biomedical Research of Salamanca (IBSAL), SACYL, Universidad de Salamanca, CSIC, 37007 Salamanca, Spain; 3Infectious Diseases Unit, Hospital Universitario de Salamanca, 37007 Salamanca, Spain; mledesma@saludcastillayleon.es; 4Department of Biomedical and Diagnosis Sciences, Faculty of Medicine, Universidad de Salamanca, 37007 Salamanca, Spain; 5Center for Research on Tropical Diseases, Universidad de Salamanca (CIETUS), 37007 Salamanca, Spain

**Keywords:** dPCR, digital PCR, infection, bacteria, virus, parasites

## Abstract

Infectious diseases account for about 3 million deaths per year. The advent of molecular techniques has led to an enormous improvement in their diagnosis, both in terms of sensitivity and specificity and in terms of the speed with which a clinically useful result can be obtained. Digital PCR, or 3rd generation PCR, is based on a series of technical modifications that result in more sensitive techniques, more resistant to the action of inhibitors and capable of direct quantification without the need for standard curves. This review presents the main applications that have been developed for the diagnosis of viral, bacterial, and parasitic infections and the potential prospects for the clinical use of this technology.

## 1. Introduction

Molecular diagnostics has revolutionised clinical microbiology in recent decades. The diagnostic techniques that have emerged from the discovery of the polymerase chain reaction (PCR) have exponentially increased the portfolio of microbiological diagnostics available in most hospitals, as well as their sensitivity, specificity, and speed. Among the several variants of PCR-based diagnostic techniques developed in recent years, digital PCR (dPCR) can bring significant improvements to the diagnosis of infectious diseases [1,2,3,4,5,6].

dPCR is a PCR that uses modified procedures, which allows it to be more accurate and sensitive than quantitative PCR (qPCR). A single reaction mixture is used for qPCR, in which the fluorescence emitted upon DNA amplification is quantified after each amplification cycle. The exact quantification of the DNA in the original sample is inferred from the comparison, during the logarithmic amplification step, between the behaviour of the test sample and a standard curve constructed from a series of calibrators that are amplified in parallel with that sample.

In contrast, dPCR subdivides, by different techniques (microwell plates, capillaries, oil emulsion), the reaction mixture into thousands of individual micro-partitions so that, under optimal conditions, each of these micro-partitions would contain a single DNA template ready for amplification. 

These micro-partitions can be obtained by an emulsion of microparticles suspended in oil (digital droplet or ddPCR) or by using microwells and microfluidic techniques. From here, an end-point PCR quantifies how many of these micro-reactors do or do not fluoresce and thus amplify, allowing the concentration of the template in the original sample to be determined. Based on this basic concept, different technologies have been developed that vary primarily in the method of generating these micro-reactors and the number of micro-reactors generated.

Conceptually, dPCR has the following advantages over qPCR: dPCR allows direct quantification of the sample without the need for calibration curves. This non-need for calibration curves is also particularly important in cases where quantified DNA is not readily available to produce calibration curves [6];dPCR allows more accurate quantification than qPCR [7];Is a more robust test in that it is resistant to many of the inhibitors that can alter qPCR results [8];Has a higher tolerance to point defects in complementarity between the template and the primers or probes, which eventually facilitates the detection of mutated subpopulations [9];Improves the comparability of results between different centres and laboratories.

Moreover, at least from a theoretical point of view, the possibilities for multiplexing dPCR are greater than for qPCR, which is more limited by the number of spectrally distinct fluorophores available [10,11].

However, dPCR also has limitations. The fact that it starts, at least for some of the available platforms, from lower sample volumes may have an obvious impact on the sensitivity of the technique.

On the other hand, the dynamic range of dPCR is also affected by the number of partitions available for analysis, so it will also be affected to a greater or lesser extent depending on the platform used.

It is a method that does not differentiate between viable and non-viable microorganisms, although this is obviously a limitation common to all the usual molecular diagnostic techniques.

We must not forget, especially if we focus on clinical applications, the speed of availability of results, which currently cannot compete with the fastest qPCR-based applications, the cost, which is also currently significantly higher than commercial qPCR techniques, and the possibility of integration into fully automated diagnostic equipment and structures that are currently barely developed for dPCR.

## 2. Virology Applications

The diagnostic applications developed earlier, once this technology was developed, were quantitative virological diagnostic techniques in relation to the advantages that this technology offers for quantification. Thus, techniques have been developed for different pathogens such as some types of polyomaviruses, HTLV-1, GB hepatitis virus, hepatitis B CCC viral DNA, Epstein–Barr virus, cytomegalovirus, SARS-CoV-2 virus [12] and others. In the case of hepatitis B, a study has shown that dPCR would have a sensitivity for hepatitis B virus genome detection in liver tissue, some ten-fold higher than the already very high sensitivity of qPCR [13].

One area in which it has provided important information is the study of HIV infection, in some respects where previously available techniques had limitations. Some publications have studied cell-associated HIV DNA by dPCR. It has been shown that ddPCR allows a much more reliable and accurate measurement of total HIV DNA and episomal 2-LTR circles in cells isolated from infected patients, apart from the mentioned advantages of not depending on external standards for quantification and of being much less conditioned by eventual mismatches of probe and primer sequences. The possibility of measuring proviral DNA much more accurately allows a much more precise knowledge of the cellular reservoir [14,15]. 

Several studies have demonstrated a higher sensitivity in the detection of HIV RNA and, as a consequence, earlier detection [16,17].

dPCR has also been used to predict and follow the course of the disease by measuring the number of copies of the gene encoding the chemokine CCL4L, which encodes the ligand for CCR5 and thus acts as a suppressor of infection [18].

These techniques allow a more comprehensive view of the different HIV reservoirs and may be essential in the better understanding of the infection and in the development of new drugs aimed at eradicating the infection [6].

Another area where dPCR makes an important contribution is the detection of the integrated genome of different viruses. Thus, it has served to make important advances in the detection of inherited chromosomally integrated (ICI) human herpesvirus-6 (HHV-6). In HHV-6 infection, integration of the virus into germ cells results in the offspring of infected persons having ICI-HHV-6 in all cells of the body. dPCR allows the study of the viral DNA/human DNA ratio in these cases with remarkable accuracy [19,20]. 

A recent study shows that dPCR would have a limit of quantification for both cytomegalovirus and Epstein–Barr virus of around 100–150 copies/mL, both in serum and plasma, with very low coefficients of variation and high comparability between different devices [21], while a retrospective study with dPCR on human adenovirus reactivation after haematopoietic stem cell transplantation [22], showed that dPCR exhibited better reproducibility and sensitivity, as well as equivalent quantifiability, compared with qPCR. dPCR detected DNA among HSCT patients in 49 (9.0%) of 545 samples, while qPCR was positive in only 11 samples (2.0%).

However, it has not been demonstrated in all cases that these technically obvious improvements lead to improvements in patient management. A study published in 2014 showed that follow-up of cytomegalovirus viremia in haematopoietic stem cell transplant patients using dPCR did not lead to faster or more appropriate treatment compared to patients monitored using qPCR [23]. 

From a technical point of view, dPCR also has some advantages. As discussed above, it is considered a more inhibition-resistant technology because, as each genome is amplified individually, even amplifications that occur with lower efficiency would be detected and considered positive. This feature may be important in some types of samples that are more prone to inhibition, such as faecal samples [24].

The same technical basis explains why it is a more reliable technique for quantifying viruses that are prone to show significant genetic heterogeneity. By amplifying each genome individually, even those genomes with suboptimal complementarity with respect to the primers or probes would give a certain level of signal, albeit less intense than in those pairs with optimal complementarity, and that signal would be recorded and considered as positive. In this way, the tendency to under-quantification that can be observed, in these circumstances, in several viral infections, as has been demonstrated in the case of hepatitis and HIV viruses, could be largely avoided.

In this regard, the diagnosis of human papillomavirus (HPV) infections by qPCR may be limited by both factors. The presence of nucleotide mismatches at the primer binding sites may result in suboptimal annealing and, consequently, under-detection of some HPV types. On the other hand, when anal samples are involved, possible faecal contamination increases the risk of inhibition of the amplification reaction. A study assesses if touchdown dPCR using broad-spectrum oligonucleotides and genotype-specific probes targeting HPV L1 can be used to quantitate HPV, both in general and in anal samples [25]. The touchdown dPCR detected one copy of both HPV-16 and HPV-18 types, with no cross-reactivity to 27 other low-risk and high-risk HPV types. Testing self-collected anal samples positive for HPV16 or HPV18, and those negative for both HPV types with the touchdown dPCR assay and a commercial multiplex tr-qPCR showed high agreements between the two assays.

Similarly, its reliability has been demonstrated in the detection of both SARS-CoV-2 as a whole and of SARS-CoV-2 viral mutants that may be associated with increased antiviral resistance or atypical behaviour from the pathogenic or diagnostic point of view [6,26,27]. Thus, some studies have shown that dPCR would detect between 2.2 and 3.7 copies/reaction [27] and, compared to RT-qPCR, would increase the positive rate by 12.0% in samples from SARS-CoV-2 infected patients [27]. Additionally, dPCR detected SARS-CoV-2 in three of twenty-six specimens from close contacts that had been negative by RT-qPCR. Additionally, some studies have shown that dPCR would be able to detect subpopulations of viral mutants that do not exceed 0.1% of the total population [6].

In addition, its greater accuracy in quantification has resulted in greater comparability of results from different laboratories, as well as in greater reliability of international standards, many of which have been developed and quantified using this technique [23,28,29].

## 3. Bacteriology Applications

In the field of bacteriology, numerous resources based on dPCR have been developed for the detection and identification of pathogenic bacteria, and some have, in fact, already been validated for clinical use. 

However, in the case of dPCR applications to bacteriological diagnostics, one of the main challenges is to take advantage of its theoretical advantages over qPCR in terms of sensitivity, resistance to inhibitors and ability to detect mutants in those diagnostic areas where the performance offered by qPCR can clearly be improved.

Thus, dPCR techniques have been developed for the faecal detection of *Helicobacter pylori* as an alternative to the techniques for the detection of labelled CO_2_ in expired air and the detection of faecal *H. pylori* antigen [30] for the detection of different species of *Borrelia* [31], or for the diagnosis of *Mycobacterium tuberculosis*, both in tissue samples and at the circulatory level [32]. 

In the case of Lyme disease, serological diagnosis, which has been the reference method for years, combining enzyme immunoassay and Western blotting, has significant room for improvement, especially regarding the accuracy of the enzyme immunoassay. Conventional molecular techniques are not very reliable either, as they often show false positives in both blood and CSF. A study on the application of dPCR to the diagnosis of Lyme disease was published in 2017 with promising results [33], as it was able to detect bacterial loads as low as 10 microorganisms/sample.

A more recent study has developed a multiplex dPCR with a sensitivity of 0.2–5 genome equivalent DNA copies per microlitre for different members of the *Bartonella* and *Borrelia* genus [34]. 

Important in this regard are those infections, such as tuberculosis, in which the infected area is difficult to access, and the release of DNA from the microorganism into the circulation is too low to be detected efficiently by qPCR in blood. In fact, some studies also show promising results in the diagnosis of tuberculous meningitis, tripling the sensitivity of conventional qPCR (57.4% vs. 22.1%) with similar specificity [35]. 

Already, a paper published in 2017 showed that dPCR was able to detect 100% of the cases of both pulmonary and extrapulmonary tuberculosis, while qPCR only detected around 50% [36]. 

A recently published meta-analysis on the same topic [37], which includes 14 studies with more than 1500 participants, is less optimistic, in that it suggests a similar sensitivity between qPCR and dPCR in pulmonary tuberculosis but recognises a higher sensitivity for dPCR in cases of extrapulmonary tuberculosis.

However, a recent study on different clinical sample types, using dPCR, next-generation sequencing (NGS) and conventional commercial qPCR simultaneously, showed that the most sensitive method was dPCR (99%), over NGS (86%) and conventional qPCR (64%) [38]. 

A similar study also published recently on 74 tuberculosis patients diagnosed by culture, dPCR, qPCR, and acid-fast staining showed a sensitivity of dPCR of 100%, compared to 82.4% for qPCR and 41.9% for acid-fast bacilli (AFB) smear microscopy [39].

On the other hand, recent studies demonstrate its potential value in the quantification of resistant subpopulations of *M. tuberculosis*. A specific dPCR developed to quantify isoniazid-resistant subpopulations has demonstrated a sensitivity and sensitivity greater than 9% compared to traditional drug susceptibility testing [40].

Another recent study on bacterial meningitis demonstrates that a multiplex dPCR designed to detect eight pathogens (*Neisseria meningitidis*, *Haemophilus influenzae*, *Streptococcus pneumoniae*, *Escherichia coli*, *Staphylococcus aureus*, *Streptococcus agalactiae*, *Listeria monocytogenes* and *Salmonella enterica*) has a very high correlation with conventional culture (94.4% sensitivity, 100% specificity) [41], although the design of the panel, in terms of the microorganisms it detects, is surprising and certainly questionable for a panel designed for the diagnosis of meningitis in humans, and has the additional limitation that it does not establish comparisons with any conventional RT-PCR, which would really be the alternative in terms of technology and speed of results. Some other locations in which, for different reasons (low bacterial load, difficulty of growth in conventional culture media), both conventional bacteriological techniques and RT-PCR offer improvable results can also benefit from the characteristics of dPCR. This includes conditions such as endocarditis, osteoarticular infections, and infections such as hospital-acquired pneumonia, in which reliable quantification of microorganisms in respiratory samples such as bronchoalveolar lavage (BAL) may be important for diagnosis.

In recent years, dPCR methods have been described for the specific diagnosis of numerous microorganisms. Thus, methods for the specific diagnosis of *Veillonella* have been described [42]. In this study, the sensitivity of dPCR was 11.3 copies/μL, while the sensitivity of qPCR was 100 copies/μL, though qPCR had a wider detection range than ddPCR.

A recent study [43] describes a duplex qPCR for *Burkholderia cepacia* and *Stenotrophomonas maltophilia* directly from patient blood. The specificity of the assay was found to be 100%. The duplex dPCR assay demonstrated good repeatability and could detect as low as 5.35 copies/reaction of *B. cepacia* and 7.67 copies/reaction of *S. maltophilia*. This level of sensitivity was consistent in the simulated blood and blood bottle samples. 

Although these methods retain the advantages of this technology in terms of sensitivity, resistance to inhibitors, reliability of quantification, etc., the main problem with these methods is the real usefulness of their translation to large-scale clinical diagnosis. Since these microorganisms do not give rise to clinical pictures that allow a specific diagnostic suspicion and are not among the usual aetiological agents of frequent infectious syndromes, their inclusion in multiple diagnostic panels is questionable, in the same way, that they are not included in the diagnostic panels usually available in qPCR. This aspect of the improvement that this new technology can bring with respect to the methods currently available will be a fundamental factor in its introduction. Thus, in a recent study on the application of dPCR for the diagnosis of *Mycoplasma pneumoniae*, compared to qPCR, the limit of detection of dPCR was 2.9 copies/reaction, while that for real-time PCR was 10.8 copies/reaction. In total, over 178 clinical samples, the dPCR assay identified and differentiated 80 positive samples, whereas the real-time PCR identified 79 samples. Only one sample that tested negative in real-time PCR was positive in ddPCR, with a bacterial load of three copies/test [44]. In these circumstances, it will be necessary to carefully evaluate the specific advantages that this technology may offer compared to technology such as qPCR, which is already well established, has complete multiplex panels, is fast, highly automated, and is increasingly cheaper.

Another study reports good results for the detection of *Brucella* spp. [45]. The dPCR results showed that the limit of detection was 1.87 copies per reaction, with high repeatability. The positive rates for dPCR and qPCR were 88.5% and 75.4% among 61 serum agglutination test-positive patients. However, the most significant data are that 57.6% of suspected sero-negative samples were positive by dPCR, but only 36.3% were positive by qPCR. Therefore, dPCR could be a very useful diagnostic tool in patients with clinical suspicion of brucellosis and negative serology.

Along the same lines, a recent study demonstrates the ability of dPCR to detect and quantify DNA from *Escherichia coli*, *Klepsiella pneumoniae*, *Staphylococcus aureus* and *Enterococcus* spp. in blood samples from patients with bacteraemia, with a low detection limit of between 0.1 and 1 pg/mL [46]. This makes it possible to detect the microorganism in the blood of patients with bacteraemia in a time window of 3–4 h, without having to wait for the positivisation of the conventional blood culture. This speed of detection is obviously an important advance. However, based on this finding, and given that, in many cases, there are no clinical data that allow us to suspect the specific aetiology of a case of bacteraemia, the pending challenge is to multiplex these dPCRs in order to be able to simultaneously detect the main species that cause most bacteraemia.

A second study with a similar design [47] evaluated the effectiveness of a multiplex dPCR in detecting bacterial pathogens in the blood of COVID-19 critically ill patients. The five panels developed target simultaneously *Pseudomonas aeruginosa*, *K. pneumoniae*, *E. coli*, *Acinetobacter baumannii*, *S. aureus*, *Enterococcus* spp., *Streptococcus* spp., *Candida* spp., *coagulase-negative Staphylococcus*, *B. cepacia*, *S. maltophilia*, *Serratia marcescens*, *Proteus mirabilis*, *Enterobacter cloacae*, *Citrobacter freundii*, *Salmonella* spp., *Bacteroides fragilis*, *Morganella morganii* and seven antimicrobial resistance genes, including *bla_KPC_*, *bla_NDM_*, *bla_IMP_*, *Oxa-48*, *mecA*, *VanA*, and *VanM*. The study included RT-PCR-confirmed COVID-19 patients admitted to a hospital in China for a period of three months. Among the 200 samples included in the study, 22.5% were positive for bacterial targets using blood culture, while 56.5% were positive using the dPCR assay. The ddPCR assay outperformed blood culture in pathogen detection rate, mixed infection detection rate, and fungal detection rate. Of 113 ddPCR-positive cases, 76 were BC-negative, and an additional three were ddPCR+/BC+ but with inconsistent bacteria. Forty-eight (60.8%) of these 79 episodes fulfilled the criteria for probable BSI, while 13 (16.5%) were categorised as possible BSI. The remaining 18 cases (22.8%) were presumptive false positives. Otherwise, 11 blood culture-positive samples within the target range tested negative by ddPCR. Most of these false-negative dPCRs were observed in blood cultures positive for Gram-positive bacteria, suggesting that the reaction conditions for dPCR were still suboptimal, especially for Gram-positive bacteria.

Similarly, methods for highly sensitive detection of antimicrobial resistance genes based on dPCR are being developed. A method for screening for the presence of *vanA* and *vanB* genes in faecal samples has been developed [48]. The limit of detection was 46.9 digital copies (dcp)/mL for *vanA* and 60.8 dcp/mL for *vanB*. The assay showed good linearity between 4.7 × 10^1^ and 3.5 × 10^5^ dcp/mL (*vanA*) and 6.7 × 10^2^ and 6.7 × 10^5^ dcp/mL (*vanB*). Sensitivity was 100% for *vanA* and *vanB*, with a high positive predictive value for *vanA* (100%) but lower for *vanB* (34.6%), likely due, according to the authors, to the presence of *vanB* DNA-positive anaerobic bacteria in rectal swabs.

In the case of resistance gene detection, dPCR may have a clear utility in the detection of carriers of specific resistance genes for the purpose of greater effectiveness of isolation measures and control of the spread of antimicrobial resistance. The utility will probably be more limited in terms of its usefulness for the determination of antimicrobial resistance in specific infections, where the enormous variability of resistance mechanisms and genes makes it practically impossible to develop sufficiently comprehensive panels.

## 4. Applications in Parasitology and Protozoa

Applications of dPCR have been developed for the diagnosis of protozoa that have an important epidemiological impact in specific geographical areas, such as *Trypanosoma cruzi* [49] and *Plasmodium* spp. [50,51,52] with good results. It should be taken into account that a reliable count of the parasitisation index, in the case of *Plasmodium* spp., is important to define the appropriate treatment so dPCR can make a contribution in this respect.

In some cases, studies have also been developed to detect early genes associated with resistance to different antimalarials [53].

Some other studies have been published concerning the diagnosis of protozoa such as *Cryptosporidium* [54,55] and parasites such as *Schistosoma japonicum* [55,56,57] or *Ascaris lumbricoides* [55,58], but in none of these cases have they been shown to offer obvious advantages over conventional qPCR.

As has happened in the field of bacteriology, the group of microorganisms for whose diagnosis dPCR-based techniques have been described has been expanding. A recent study [59] shows that dPCR is capable of detecting the presence of *Demodex* spp. in 71.6% of cases of blepharitis in which this aetiology is suspected, although, at least in this study, its efficacy is not superior to that of the classic method of mite counting.

## 5. Epidemiology

One area where PCR has shown great potential is in relation to epidemiological studies. Wastewater-based epidemiology (WBE) has been used to study community circulation of individual enteric viruses and panels of respiratory diseases. In this regard, several WBE studies show the potential usefulness of dPCR. A recent two-year study in China detects SARS-CoV-2 in wastewater by dPCR and by conventional qPCR. The results indicated that both RT-dPCR and RT-qPCR are effective in detecting SARS-CoV-2 in wastewater, but RT-dPCR is capable of detecting lower concentrations of SARS-CoV-2 in wastewater [60].

Another recently published study from Korea [61] also uses dPCR for the detection of SARS-CoV-2, norovirus, hepatitis A virus and NDM-type carbapenemases in wastewater. The study showed a positive association of the SARS-CoV-2 RNA with the prevalence of COVID-19 cases and showed a promising role of community-scale monitoring of pathogens to provide considerable early signals of infection dynamics. Moreover, the study shows a high capacity of this technology to detect the presence of antibiotic resistance genes, which in turn may allow for monitoring and probably predict the level of diffusion of these resistance markers.

A third WBE study [62] was carried out at two wastewater treatment plants located in California, United States, using dPCR. This study measured concentrations of human adenovirus group F, enteroviruses, norovirus genogroups I and II, and rotavirus nucleic acids in wastewater solids for 26 months (*n* = 459 samples). They detected all viral targets in wastewater solids, human adenovirus group F and norovirus GII nucleic acids being detected at the highest concentrations and rotavirus RNA at the lowest concentrations. dPCR was shown as a sensitive detection method for enteric virus targets in WBE programmes. dPCR thus becomes a reliable methodology for WBE that can facilitate decision-making aimed at acquiring a better knowledge of enteric diseases and reducing their transmission.

However, other studies on the presence of SARS-CoV-2 in wastewater for epidemiological purposes [63] confirm excellent sensitivity and specificity for both qPCR and dPCR, but without significant differences in both methods in terms of limits of detection and limits of quantification, which move in the same orders of magnitude, nor in terms of the number of positive samples and the number of quantifiable samples. Therefore, further studies will probably be necessary to corroborate to what extent these theoretical advantages of dPCR, in terms of sensitivity, inhibitor resistance, etc., are confirmed in practice.

## 6. Conclusions

Being a PCR modality, its possible applications are as wide as those already demonstrated for other modalities, such as conventional qualitative PCR or later qPCR. Probably the main challenge at this time is to determine in which areas this PCR modality, due to its specific characteristics, will represent a significant advance with respect to the technologies already developed and in use and to what extent these advantages will be a contribution of sufficient importance to displace PCR modalities already adapted to specific clinical diagnostic needs (robustness of both equipment and procedures, speed of response, automation, multiplexing possibilities, etc.).

Probably, at this moment, its main areas of development could be three: molecular diagnostics in those sample types where amplification inhibition could be a determining problem; those situations where very high quantification reliability is required; and, finally, those clinical situations where it is foreseeable that the diagnosis could benefit from its higher sensitivity.

It should also be taken into account that dPCR is in the early stages of its development, at least as far as its clinical applications are concerned, and that it is foreseeable that it will evolve towards applications that are more adapted to healthcare needs as occurred with qPCR, in terms of systems of greater simplicity, shorter response times, shorter hands-on time and with multiplexing panels adjusted to clinical needs, and other technical advances that are beginning to emerge [64]. In this regard, some interesting initiatives have been developed recently, such as a portable and fully integrated lab-on-a-disc device based on dPCR for quantitively screening infectious disease agents [65]. This device integrates microfluidics chips, an oil-based heat exchanger, and a transmitted-light fluorescent imaging system; thus, droplet generation, PCR thermocycling, and analysis can be achieved in a single device, with a lowest detected concentration of 20.24 copies/µL for several viruses. Obviously, this is not an optimal sensitivity for a dPCR, but it illustrates the trend towards the development of simple, portable, miniaturised equipment capable of simultaneously detecting several pathogens that can be used even outside the clinical laboratory setting.

A recent study also describes the performance of a multiplex dPCR, which simultaneously detects SARS-CoV-2, influenza A viruses, enteroviruses and noroviruses of genogroups I and II. The study was not carried out on clinical samples but on wastewater, but the results are promising [66].

It should also be considered that, simultaneously, other molecular diagnostic resources, mainly NGS and qPCR, are evolving in the same direction, from slow, laborious and complex techniques to formats that are much more adapted to clinical use (some platforms based on digital microfluidics offers a high degree of automation in ultrafast, 3.7–5 min PCR, with a detection sensitivity comparable to conventional PCR [67]), including a very significant reduction in their costs, and that could represent an important competition for dPCR in many areas, including the diagnosis of infectious diseases.

## Data Availability

Not applicable.

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
