# Peer review of "Present and Future Applications of Digital PCR in Infectious Diseases Diagnosis"

_diagnostics, 2024, doi:10.3390/diagnostics14090931_

Round 1
Reviewer 1 Report
Comments and Suggestions for Authors
1. Line 58-62 are disorganized.
2. In line 107, 'development of new drug' is mentioned, would you please add a reference?
3. In line 123-127, you mentioned 'dPCR did not lead to faster or more appropriate treatment compared to patients monitored using qPCR'. While the paragraphs before and after this one are all about the advantages of dPCR,what is the purpose of this paragraph.
4. In line 177, CO2 should be CO2
5. In line 208, does TB mean tuberculosis? The abbreviation needs to be stated after the full name appears.
6. In line 303, what does 'digital copies' mean?
7. In line 398, the concentration 20.24 copies/μL is very high and does not show the advantage of dPCR, it is recommended to look for a lower example. The advantage of digital pcr is that the detection sensitivity is very high, and it can detect a single molecule. At present, the commercial qPCR detection of COVID-19 can reach 200 copies/ml.
8. Many of the references are too old, especially in 'Virology Applications', and need to be updated.
Author Response
REVIEWER #1.
- Line 58-62 are disorganized.
Answer. Thank you for your comment. We have structured this paragraph, concerning the overall advantages of dPCR, into 5 points for clarity.
- In line 107, 'development of new drug' is mentioned, would you please add a reference?.
Answer. We have added a reference for this statement.
- In line 123-127, you mentioned 'dPCR did not lead to faster or more appropriate treatment compared to patients monitored using qPCR'. While the paragraphs before and after this one are all about the advantages of dPCR,what is the purpose of this paragraph.
Answer. What we mean in this paragraph is that, although dPCR provides obvious technical improvements (greater sensitivity, greater robustness of the data), this does not always have a direct impact on favorable changes in the diagnosis and management of patients. In the publication to which we refer, technical improvements do not result in faster or more effective treatment. Therefore, it is necessary to assess the contribution of this technology in each case to determine whether its implementation is appropriate in that particular case. We have modified the wording of the paragraph to make it easier to understand.
- line 177, CO2 should be CO2
Answer. The error has been modified.
- In line 208, does TB mean tuberculosis? The abbreviation needs to be stated after the full name appears.
Answer. Since it is not a term that is used repeatedly in the text, we have chosen to use the full name, without abbreviations. As the reviewer correctly surmised, the abbreviation referred to tuberculosis.
- In line 303, what does 'digital copies' mean?
Answer. What the authors mean by the term "digital copies/ml" is that it is a copy number obtained by absolute copy number counting by digital PCR, and not an extrapolation from standard curves.
- In line 398, the concentration 20.24 copies/μL is very high and does not show the advantage of dPCR, it is recommended to look for a lower example. The advantage of digital pcr is that the detection sensitivity is very high, and it can detect a single molecule. At present, the commercial qPCR detection of COVID-19 can reach 200 copies/ml.
REV. 2
Comments and Suggestions for Authors
I see not major problems. Two comments only. In case of culturable bacteria how dPCR agrees with bacterial counts by plating on the Petri dishes? What's a costs difference?
Answer. Thank you for your comment. Indeed, it is not the most sensitive digital PCR system available. We mentioned it in this paragraph as an example of the trend towards the development of more compact and miniaturized equipment, easy to use and with fast results, more in line with the "point of care" devices that is being used in other diagnostic fields, including conventional qPCR techniques. We have modified the paragraph for a better understanding.
The second comment is common for all methods employing so-called limiting dilution concept. Is calculation of dPCR comparable with L-calc or ELDA calculators. It is worth mentioning anyway. For those methods it is common assumption that cells you count do not clump. Is there any way to discover possible clumps of cells in dRCR?
Answer. Thank you for your comment. Indeed, it is not the most sensitive digital PCR system available. We mentioned it in this paragraph as an example of the trend towards the development of more compact and miniaturized equipment, easy to use and with fast results, more in line with the "point of care" devices that is being used in other diagnostic fields, including conventional qPCR techniques. We have modified the paragraph for a better understanding.

Reviewer 2 Report
Comments and Suggestions for Authors
I see not major problems. Two comments only. In case of culturable bacteria how dPCR agrees with bacterial counts by plating on the Petri dishes? What's a costs difference?
The second comment is common for all methods employing so-called limiting dilution concept. Is calculation of dPCR comparable with L-calc or ELDA calculators. It is worth mentioning anyway. For those methods it is common assumption that cells you count do not clump. Is there any way to discover possible clumps of cells in dRCR?
Comments on the Quality of English Languageno problems detected
Author Response
REVIEWER #2.
- In case of culturable bacteria, how dPCR agrees with bacterial counts by plating on Petri dishes?
Answer. The comparison between the count obtained by the conventional colony count method and dPCR can be highly variable, especially under clinical conditions. It should be taken into account that this variability can be conditioned by numerous factors, such as the number of genomes belonging to non-viable bacteria, especially if there has been antibiotic treatment, which would be detected by dPCR but would not lead to colony formation. Also, the correlation between the number of colony forming units and colonies actually formed, which in many cases does not have a 1:1 ratio. Still, some experimental studies, under controlled conditions, (Barrett-Manako K, et al., Real-Time PCR and Droplet Digital PCR Are Accurate and Reliable Methods To Quantify Pseudomonas syringae pv. actinidiae Biovar 3 in Kiwifruit Infected Plantlets. Plant Dis. 2021; 105: 1748-1757) show a high correlation between the two methods (r>0.85).
- What’s a costs difference.
Answer. At present, the cost of dPCR is higher than that of conventional qPCR, and obviously much higher than that of classical bacterial counting. Therefore, in my opinion, it is not a method to be used universally and systematically, but only in those circumstances in which some of its characteristics, such as its very high sensitivity or its capacity to carry out a reliable count of mutants and subpopulations, are of real clinical importance, as may be the case of counting viral subpopulations, or the diagnosis of infectious conditions with a low bacterial load in blood or tissues, as could be the case, for example, of endocarditis.
- Is there any way to discover possible clups of cells in dPCR?
Answer. Indeed, the possibility of clumping is one of the problems that can bias bacterial counts downward, when considering a clump as a single microorganism. However, this has not been an effect that has been observed when counting with genome fragments or amplicons, so in cases where bacterial clumps can occur, counting based on genome quantification would probably be more reliable.